# Sea Anemone Kunitz-Type Peptides Demonstrate Neuroprotective Activity in the 6-Hydroxydopamine Induced Neurotoxicity Model

**DOI:** 10.3390/biomedicines9030283

**Published:** 2021-03-10

**Authors:** Oksana Sintsova, Irina Gladkikh, Margarita Monastyrnaya, Valentin Tabakmakher, Ekaterina Yurchenko, Ekaterina Menchinskaya, Evgeny Pislyagin, Yaroslav Andreev, Sergey Kozlov, Steve Peigneur, Jan Tytgat, Dmitry Aminin, Emma Kozlovskaya, Elena Leychenko

**Affiliations:** 1G.B. Elyakov Pacific Institute of Bioorganic Chemistry, Far Eastern Branch, Russian Academy of Sciences, 159 Pr. 100 let Vladivostoku, 690022 Vladivostok, Russia; sintsova0@gmail.com (O.S.); irinagladkikh@gmail.com (I.G.); rita1950@mail.ru (M.M.); eyurch@piboc.dvo.ru (E.Y.); ekaterinamenchinskaya@gmail.com (E.M.); pislyagin_ea@piboc.dvo.ru (E.P.); daminin@piboc.dvo.ru (D.A.); kozempa@mail.ru (E.K.); 2Shemyakin-Ovchinnikov Institute of Bioorganic Chemistry, Russian Academy of Sciences, ul. Miklukho-Maklaya 16/10, 117997 Moscow, Russia; tabval@yandex.ru (V.T.); shifter2007@gmail.com (Y.A.); serg@ibch.ru (S.K.); 3Institute of Molecular Medicine, Sechenov First Moscow State Medical University, Trubetskaya str. 8, bld. 2, 119991 Moscow, Russia; 4Toxicology and Pharmacology, University of Leuven (KU Leuven), Campus Gasthuisberg O&N2, Herestraat 49, P.O. Box 922, B-3000 Leuven, Belgium; steve.peigneur@kuleuven.be (S.P.); jan.tytgat@kuleuven.be (J.T.); 5Department of Biomedical Science and Environmental Biology, Kaohsiung Medical University, 100 Shih-Chuan 1st Road, Kaohsiung 80708, Taiwan

**Keywords:** sea anemones, Kunitz-type proteinase inhibitors, TRPV1, Kv, recombinant peptides, neurotoxicity, 6-OHDA, ROS, DPPH, cytoprotection

## Abstract

Kunitz-type peptides from venomous animals have been known to inhibit different proteinases and also to modulate ion channels and receptors, demonstrating analgesic, anti-inflammatory, anti-histamine and many other biological activities. At present, there is evidence of their neuroprotective effects. We have studied eight Kunitz-type peptides of the sea anemone *Heteractis crispa* to find molecules with cytoprotective activity in the 6-OHDA-induced neurotoxicity model on neuroblastoma Neuro-2a cells. It has been shown that only five peptides significantly increase the viability of neuronal cells treated with 6-OHDA. The TRPV1 channel blocker, HCRG21, has revealed the neuroprotective effect that could be indirect evidence of TRPV1 involvement in the disorders associated with neurodegeneration. The pre-incubation of Neuro-2a cells with HCRG21 followed by 6-OHDA treatment has resulted in a prominent reduction in ROS production compared the untreated cells. It is possible that the observed effect is due to the ability of the peptide act as an efficient free-radical scavenger. One more leader peptide, InhVJ, has shown a neuroprotective activity and has been studied at concentrations of 0.01–10.0 µM. The target of InhVJ is still unknown, but it was the best of all eight homologous peptides in an absolute cell viability increment on 38% of the control in the 6-OHDA-induced neurotoxicity model. The targets of the other three active peptides remain unknown.

## 1. Introduction

Peptides with Kunitz/BPTI domains are common throughout all taxa (from bacteria to mammals). In vertebrates, they participate in the regulation of inflammation, blood coagulation, and digestion, while in invertebrates, they are involved in the development of the reproductive system, participation in allergic and immune reactions, regulation of collagen synthesis, and other physiological processes [1]. The mechanism of their action is primarily associated with the inhibition of proteinases (mainly serine proteinases); however, some of them have other targets. It has been found that some Kunitz-type peptides from venomous animals modulate type-2 vasopressin receptors and integrins or TRPV1, Kv, Nav, Cav, and ASIC channels [2,3,4,5,6,7,8,9,10,11], demonstrating analgesic, anti-inflammatory, antihistamine activity and many other pharmacological effects as well [10,12,13,14,15,16]. Moreover, Kunitz-type peptides from snakes, spiders, scorpions, cone snails, and sea anemones have been found to produce Kv blockage and exhibit toxic effects. The 3D structure of the all known peptides is highly conserved; it includes Kunitz/BPTI fold (length ~ 60 amino acid residues), stabilized by three disulfide bonds (Cys^I^–Cys^VI^, Cys^II^–Cys^IV^, Cys^III^–Cys^V^). Generally the peptides of this structural family form combinatorial libraries [17,18,19,20], the members of which have up to 10 point mutations in the sequence that do not affect the spatial structure of the molecule but lead to the diversification of biological activities toward various targets [21].

Recently, we have investigated combinatorial libraries of Kunitz-type peptides of the sea anemones *Heteractis crispa* and *Heteractis magnifica* [18,22]. To study their biological activity, we have isolated some of these peptides (APHC1-3, InhVJ, RmInI, RmInII, HCRG1, and HCRG2) from the bodies of sea anemones while the others (HCGS1.10, HCGS1.19, HCGS1.20, HCGS1.36, HCRG21, HCIQ2c1, HCIQ4c7, HMIQ3c1) have been obtained as recombinant analogs only [10,12,13,14,15,23,24,25,26]. All the investigated peptides inhibit trypsin (K_i_ from 2.1 × 10^−8^ up to 1 × 10^−6^ M); some of them are able to interact with serine proteinases involving in coagulation, blood pressure control and inflammation, such as kallikrein, cathepsin G, and elastase [27,28]. In contrast to potent serine proteinase inhibitors, the proteinase inhibitory activity of these peptides is lower, which suggests that their main function should be determined for other targets. This is indirectly confirmed by the fact that we have detected anti-inflammatory activity of RmInI and RmInII in vivo, which was manifested in the weakening of inflammation in response to histamine injection, and demonstrated that the HCGS1.10, HCGS1.19, HCGS1.20, and HCGS1.36 peptides suppressed the increase of Ca^2+^ response under the influence of histamine in vitro [15,24], and HCGS1.20 reduced the content of nitric oxide by 25% in macrophages treated by lipopolysaccharide [24]. 

In addition, for some of them, cellular targets have already been found that allow explaining their biological effects. It has been found that TRPV1 is the target of APHC1-3 (partial inhibition up to 50%) and HCRG21 (full inhibition) resulting in analgesic and anti-inflammatory effect in mice [10,12,29,30], whereas Kv channels are the targets of HCRG1 and HCRG2 that block them and demonstrate anti-inflammatory activity reducing expression level of TNF-α, IL-6, and proIL-1β in macrophages [14], as well as suppressing TNF-α production in acute carrageenan-induced edema model [31]. 

It is well known, that chronic inflammation underlies many serious diseases including neurodegenerative ones, such as Alzheimer’s and Parkinson’s. It was found *H. magnifica* Kunitz-type peptide, HMIQ3c1, exhibited neuroprotective activity in Alzheimer’s disease model, increasing the viability of Neuro-2a murine neuroblastoma cells in the presence of β-amyloid [26]. Kunitz-type peptide PcKuz3 from zoanthid *Palythoa caribaeorum* suppressed the 6-OHDA-induced neurotoxicity (animal Parkinson’s disease model) expressed in violation of the locomotor behavior of zebrafish *Danio rerio* [32]. More recently, it has been shown that *H. crispa* peptide HCIQ2c1 demonstrated neuroprotective activity and suppression of ROS synthesis in 6-OHDA-treated neuroblastoma Neuro-2a cells [27]. The mechanism of action has not yet been elucidated, but it was found that HCIQ2c1 inhibits different proteinases (trypsin, α-chymotrypsin, kallikrein, neutrophil elastase, and cathepsin G) and does not block Kv channels [27]. 

In continuation of studies of the neuroprotective potential of Kunitz-type peptides of the sea anemone *H. crispa*, the influence of eight peptides on viability and ROS level in the 6-OHDA-treated neuroblastoma Neuro-2a cells as well as their DPPH radical scavenging activity in cell free assay were conducted. Peptides with unidentified targets were subjected to electrophysiological assay on TRPV1 and eight isoforms of Kv channels expressed in *Xenopus laevis* oocytes.

## 2. Materials and Methods

### 2.1. Isolation of Peptides from Sea Anemone Heteractis Crispa

The peptides InhVJ, HCRG1, and HCRG2 were isolated from the sea anemone *Heteractis crispa* according to [14,23]. In brief, InhVJ was purified from a 70%-ethanol extract by the scheme that included a hydrophobic chromatography on Polychrome-1 (powdered Teflon, Biolar, Olaine, Latvia) column (4.8 × 95 cm), gel filtration chromatography on Bio-gel P-4, and ion-exchange chromatography on SP-Sephadex G-25, and two steps of RP-HPLC on Nucleosil C_18_ column (4.6 × 250 mm) (Sigma Aldrich, St. Louis, MO, USA). HCRG1 and HCRG2 were precipitated from a water extract with 80% acetone, next gel filtration chromatography on an Akrilex P-4 column was carried out, followed by cation-exchange chromatography on a CM-32 cellulose column, with a final purification step using a RP-HPLC Nucleosil C_18_ column (4.6 × 250 mm) (Sigma Aldrich, St. Louis, MO, USA).

### 2.2. Production of Recombinant Peptides

The following reagents were used: Ni-NTA-agarose (Qiagen, Hilden, Germany), components of bacterial culture media (Difco, Sparks, MD, USA), BL21 (DE3) *Escherichia coli* (Novagen, Darmstadt, Germany). The expression construction pET32b(+) with target genes (*hcgs1.10, hcgs1.19, hcgs1.20, hcgs1.36, hcrg21*) were obtained previously [10,15,24]. Constructions containing *hcgs1.19, hcgs1.20, hcgs1.36* were made by cloning of natural gene from cDNA library; *hcgs1.10* and *hcrg21* by cloning of synthetic gene optimized for expression in *E. coli*. The construction pET32b(+)/*inhvj* was synthesized by JSC Eurogen (Russia). The expression constructions on the base of pET32b(+) with target genes were used for transformation of BL21(DE3) *E. coli* cells by electroporation on a Multiporator (Eppendorf, Hamburg, Germany) device. All expression and purification procedures were carried out as described [10,33]. The RP-HPLC was carried out on column Jupiter C_4_ (10 × 250 mm) (Phenomenex, Torrance, CA, USA). The solvents A and B were 0.1% TFA in water and in acetonitrile, respectively. The chromatographic runs were performed using a 0–70% gradient of solvent B over 70 min at a flow rate of 3 mL/min. UV detection was monitored at 214 nm. The yields of the recombinant peptides were: 4.27 of HCGS1.19, 4.59 of HCGS1.36, 5.5 of HCRG21, 8.19 of HCGS1.20, 9.38 of HCGS1.10, and 18.5 of InhVJ measured in mg/L cell culture (OD A600 = 0.8).

### 2.3. Mass Spectrometry Analysis

MALDI-TOF MS spectra of peptide fractions after RP-HPLC were recorded using an Ultra Flex III MALDI-TOF/TOF mass spectrometer (Bruker, Bremen, Germany) with a nitrogen laser (Smart Beam, 355 nm), reflector and potential LIFT tandem modes of operation. Sinapinic acid was used as a matrix.

### 2.4. Amino Acid Sequence Determination

Recombinant peptides HCGS1.10, HCGS1.19, HCGS1.20, HCGS1.36, HCRG21, and InhVJ were reduced and alkylated with 4-vinylpyridine as described in [14]. Alkylated peptides were desalted by RP-HPLC using a Nucleosil C_18_ column (4.6 × 250 mm) (Sigma Aldrich, USA). The solvents A and B were 0.1% TFA in water and in acetonitrile, respectively. The chromatographic runs were performed using a 10–70% gradient of solvent B over 160 min at a flow rate of 0.5 mL/min. UV detection was monitored at 214 nm.

The N-terminal amino acid sequences were determined on an automated sequencer protein Procise 492 Clc (Applied Biosystems, Foster City, CA, USA).

### 2.5. CD Spectroscopy

Circular dichroism spectra were recorded on Chirascan-plus CD spectropolarimeter (Applied Photophysics, Leatherhead, UK) in quartz cells with an optical path length of 0.1 cm and 1 cm for the peptide and the aromatic region spectrum respectively. The cuvette with the solution of the inhibitor in a 0.01 M sodium phosphate buffer was thermostated at a given temperature for 20–25 min before recording the CD spectrum. The content of secondary structure elements of proteins was calculated by the Provenzer-Glockner method [34], using advanced Provencher calculation programs from the CDPro software package [35].

### 2.6. Trypsin Inhibition Constant Determination

The trypsin inhibitory constant was determined by Dixon method [36] as described [15,24] using N-α-benzoyl-d,l-arginine p-nitroanilide (BAPNA) as a substrate, measuring the absorbance of the resulting p-nitroaniline at 410 nm using xMark plate reader (BioRad, Hercules, CA, USA). The constant was calculated from the data of three parallel experiments, the limits of the calculated error were 0.1–0.5%.

### 2.7. Cell Culture

Neuro-2a neuroblastoma cells (ATCC^®^ CCL-131™) were obtained from American Type Culture Collection (Manassas, VA, USA). The cells were cultivated in DMEM medium (BioloT, St. Petersburg, Russia) supplemented with 10% of fetal bovine serum (BioloT, Russia) and 1% penicillin/streptomycin (BioloT, St. Petersburg, Russia) at 37 °C and 5% CO_2_. Initially, cells were incubated in cultural flasks until sub-confluent (~80%). For testing, Neuro-2a cells were seeded in 96-well plates and experiments were started after 24 h [37].

### 2.8. Cell Viability Assay

Peptides at different concentrations were added in Neuro-2a cell monolayer (1 × 10^4^ cells per well of 96-well plate). Then cells were incubated for 24 h at 37 °C and 5% CO_2_. Thereafter, the medium was replaced with clean serum-free medium, and the number of viable cells was determined by MTT assay. MTT reagent (3-(4,5-dime thylthiazol-2-yl)-2,5-diphenyltetrazolium bromide) (Sigma-Aldrich, St. Louis, MO, USA) was added for 4 h, then the medium was collected and whole cells were lysed in 100% dimethylsulfoxide (DMSO), the absorbance was measured at 570 nm using Multiscan FC (Thermo Fisher Scientific, Waltham, MA, USA) [38]. The cytotoxic activity of compounds was calculated as the concentration of 50% metabolic activity cell inhibition (IC_50_). Results are expressed as the mean ± SEM of three independent replicates.

### 2.9. 6-Hydroxydopamine-Induced Cytotoxicity

Neuro-2a cells were seeded at 1 × 10^4^ cells/well in 96-well plates for studying of neuroprotective activity of peptides. The tested peptides were added to the cell monolayer and cells were incubated at 37 °C for 1 h. Then, the cells were exposed to 6-OHDA (Sigma-Aldrich, St. Louis, MO, USA) at 25 µM for 24 h at 37 °C with 5% CO_2_. Thereafter, the medium was replaced with clean serum-free medium, and the number of viable cells was determined by MTT assay [38]. Neuro-2a cells were cultured without test compound or neurotoxins as the control group. Results were expressed in percentage of control as the mean ± SEM of three independent replicates.

### 2.10. Evaluation of Intracellular ROS Level

The level of intracellular ROS was measured using the ROS sensitive dye, 2,7-dichloro-fluorescein diacetate (DCFH-DA), as a probe. In brief, Neuro-2a cells were seeded in 96-well plates at 1 × 10^4^ cells/well, culturing in the presence or absence of various concentrations of tested peptides for 1 h, and then 6-OHDA (25 µM) was added in each well for 30 min at 37 °C with 5% CO_2_. Then cells were washed three times with PBS (BioloT, St. Petersburg, Russia) and incubated with a final concentration of 10 µM DCFH-DA (Molecular Probes, Eugene, OR, USA) for 30 min at 37 °C in the dark. After incubation, cells were washed three times and with a free-serum medium. The fluorescence of 2,7-dichlorofluorescein (DCF) was detected with PHERAstar FS fluorescent plate reader (BMG Labtech, Ortenberg, Germany) at λex = 485 nm and λem = 518 nm. Data were processed using MARS Data Analysis V3.01 R2 software (BMG Labtech, Ortenberg, Germany). The cytoprotective effect of peptides and the increase in cell viability under their action was calculated as a percentage, taking the cell viability treated with 6-OHDA as 100%.

### 2.11. DPPH Radical Scavenging Assay

The 2,2-diphenyl-1-picryl-hydrazyl-hydrate (DPPH) radical scavenging activity was tested as described previously [39]. The compound solutions (120 µL) were dispensed into wells of a 96-well microplate. The DPPH (Sigma-Aldrich, Germany) was dissolved in MeOH at concentration of 7.5 µM and the solution (30 µL) was added to each well. The concentrations of tested compounds in the mixtures were 10 µM. The mixtures were shaken and left for 30 min. The absorbance of the resulting solutions was measured at λ = 520 nm with a MultiscanFC plate reader (Thermo Fisher Scientific, Waltham, MA, USA). The scavenging of the DPPH radical in comparison with control (MeOH) (%) was calculated for each compound. Ascorbic acid at 10 µM was used as positive control.

### 2.12. Bioassays Data Evaluation

All data were obtained in three independent replicates and calculated values were expressed as mean ± standard error of the mean (SEM). The Student’s *t*-test was performed using SigmaPlot 14.0 (Systat Software Inc., San Jose, CA, USA) to determine statistical significance.

### 2.13. Expression of Voltage-Gated Ion Channels in Xenopus Laevis Oocytes

For the expression of rKv1.1, hKv1.2, hKv1.3, rKv1.4, rKv1.5, rKv1.6, *Shaker* IR, hERG and TRPV1 in *Xenopus laevis* oocytes, the linearized plasmids were transcribed using the T7 or SP6 mMESSAGE-mMACHINE transcription kit (Ambion, Austin, TX, USA). The harvesting of stage V–VI oocytes from anaesthetized female *X. laevis* frog was as previously described [40,41]. Oocytes were injected with 50 nL of cRNA at a concentration of 1 ng/nL using a micro-injector (Drummond Scientific, Broomall, PA, USA). The oocytes were incubated in a solution containing (in mM): NaCl, 96; KCl, 2; CaCl_2_, 1.8; MgCl_2_, 2; and HEPES, 5 (pH 7.4), supplemented with 50 mg/L gentamicin sulfate.

### 2.14. Electrophysiological Studies

The physiological activity in oocytes expressing heterologously the voltage-gated ion channel proteins was determined by the two-electrode voltage-clamp technique, using a Geneclamp 500 amplifier (Molecular Devices, Austin, TX, United States) controlled by the pClamp database system (Axon Instruments, Union City, CA, United States). The measurements were performed at room temperature (18–22 °C). Whole-cell currents were recorded 1–4 days after the mRNA injection. The electrode resistance was 0.7–1.5 MΩ. The signal was amplified, preliminarily filtered by the amplifier embedded four-polar Besselian filter (cutoff frequency 500 Hz) after digitization of the signal at 2000 Hz. Recordings obtained before the activation of the examined currents were used for subtraction of the capacitive and leakage current. The cells were kept at a holding potential of -90 mV. The membrane potential was depolarized to 0 mV for 250 ms with a subsequent pulse to −50 mV for 250 ms in the case of the Kv1.1–Kv1.6 and *Shaker* channels. Current traces of hERG channels were elicited by applying a +40 mV pre-pulse for 2 s followed by a step to −120 mV for 2 s. TRPV1 currents were measured in ND96 solution using a protocol of −90 mV during 400 s. The recording chamber was perfused at a rate of 2 mL/min with an ND96 solution. Capsaicin (2 µM) was used as an agonist and capsazepine (10 µM) as an antagonist of TRPV1. Capsaicin and capsazepine were purchased from Sigma. For statistical analysis, the Student’s coefficient (*p* < 0.05) was used. All the results were obtained from at least three independent experiments (*n* ≥ 3) and are expressed as mean value ± standard error. The use of the *X. laevis* animals was in accordance with the license number LA1210239 of the Laboratory of Toxicology & Pharmacology, University of Leuven (Belgium). The use of *X. laevis* was approved by the Ethical Committee for animal experiments of the University of Leuven (P186/2019). All animal care and experimental procedures agreed with the guidelines of the European Convention for the protection of vertebrate animals used for experimental and other scientific purposes (Strasbourg, 18.III.1986).

## 3. Results

### 3.1. Production and Characterization of the Peptides 

To identify a possible neuroprotective effect, eight Kunitz-type peptides of *H. crispa* were obtained. Two of them, HCRG1 and HCRG2, were isolated previously from a water extract of the sea anemone according to a previously developed scheme, including precipitation of peptides with 80% acetone, gel filtration chromatography on an Akrilex P-4 column, cation-exchange chromatography on a CM-32 cellulose column, and a final purification step using a RP-HPLC Nucleosil C_18_ column [14]. The molecular mass of the peptides according to MALDI-TOF MS were 6148 and 6196 Da. The other peptides, HCGS1.10, HCGS1.19, HCGS1.20, HCGS1.36, HCRG21, and InhVJ were produced by the *Escherichia coli* expression system using recombinant plasmids created on the base of vector pET32b(+), which ensures high yields of peptides containing disulfide bonds in native conformation due to expression of target peptides as thioredoxin fusion proteins. 

Gathering of the target peptides was carried out in previously selected conditions using Ni^2+^-affinity chromatography and RP-HPLC [10,24]. The retention times of the peptides HCGS1.10, HCGS1.19, HCGS1.20, HCGS1.36, and HCRG21 were identical to those described previously [10,15,24]. The N-terminal amino acid sequence of HCGS1.10, HCGS1.19, HCGS1.20, HCGS1.36, HCRG21, and InhVJ (15 a.a. residues) was determined by the Edman automatic sequencing and appeared to be identical to that evaluated based on gene sequence. The retention time of recombinant InhVJ obtained for the first time in this work was 29 min in conditions described in Figure 1. Identity check by analytical RP-HPLC for a natural and recombinant InhVJ mixture showed one symmetrical peak on 31 min for Nucleosil C_18_ column (Sigma Aldrich, USA) (Appendix A). The measured molecular weight of recombinant InhVJ was 6107.9 Da (Figure 2), which is in good agreement with the measured one for natural InhVJ and calculated (6106.4 Da) (Table 1). 

The molecular weight of the other recombinant peptides measured by MALDI-TOF MS technique was in good accordance with the calculated values (Appendix A) (Table 1). The yields of the recombinant peptides varied from 4.29 to 18.5 mg/L of the cell culture (OD A_600_ = 0.8) [10,15,24,25]. 

### 3.2. CD Spectra of the Recombinant InhVJ

CD spectra of the recombinant InhVJ in the near (230–310 nm) and far (190–240 nm) UV regions revealed a secondary structure similar to the native peptide (Figure 3). The spectrum in the near UV region (Figure 3a) has a clearly defined structure with negative bands at 285 and 275 nm that correspond to tyrosine residues, and at 268 and 262 nm corresponding to the residues of phenylalanine. The negative band of high ellipticity at 240 nm, apparently due to disulfide bonds, also is present the spectrum. The pronounced fine structure of the spectrum indicates a significant asymmetry in the surrounding of the aromatic amino acid residues, their rigid fixation in the molecule, and the highly organized tertiary structure of the recombinant peptide. The CD spectrum in the far UV region (Figure 3b), the absorption region of peptide bonds, is characterized by a minimum at 202 nm and a maximum at 193 nm. In the 225–230 nm region, a distinct shoulder is observed on the spectrum curve due to the contribution of the disulfide groups absorption.

According to the calculated data for the CD Pro program [35], the recombinant InhVJ contains 21.1% of helices (12.4%—α-helix, 8.7%—3_10_ helix), 34.6% β-structure (10.1%—β-sheets, 24.5%—turns), and 44.3% of the disordered structure (Table 2). Compared to the structure of the native peptide, the content of α-helices of the recombinant peptide is increased, while the content of β-sheets is reduced, but the same total content of turns and disordered structure is maintained. A similar pattern was observed for recombinant analogs of sea anemone α-amylase inhibitors, which did not affect their function [42].

The inhibition constant (K_i_) of recombinant InhVJ for trypsin was determined using Dixon’s method [36] and proved to be 7.8 × 10^−8^ M (Table 1) which is almost identical to the constant of the native peptide (7.38 × 10^−8^ M) [23]. In summary, recombinant InhVJ has a correct spatial organization, as well as the trypsin inhibition constant [23], which suggests proper folding and functionality.

### 3.3. Neuroprotective Activity of Kunitz-Type Peptides and Influence on Reactive Oxygen Species Production

We examined the effect of peptides set on cell viability in 6-OHDA-induced neurotoxicity model. The peptides were nontoxic to Neuro-2a cells at a concentration 10 µM. The viability of Neuro-2a cells treated with 6-OHDA was 42.6% versus control (Figure 4). Pre-treatment of cells with peptides for 1 h before 6-OHDA adding provided an increase of cell viability with different statistical confidence. Peptides HCGS1.10 and HCGS1.20 increased 6-OHDA-treated cell viability by 31.7% and 20.9% versus 6-OHDA-treated cells, respectively, with a p-value less than 0.05. Peptides HCGS1.36, HCRG21, and InhVJ increased 6-OHDA-treated cell viability by 22.8%, 23.2%, and 38% respectively with p value less than 0.01. HCRG1 and HCRG2 did not show significant improvement in this assay (Figure 4).

It is believed that the cell death in the 6-OHDA-induced neurotoxicity model is a result of dramatically increased reactive oxygen species (ROS) content [43]. For this reason, the Kunitz-type peptides have been tested for their ability to influence ROS production in 6-OHDA-treated Neuro-2a cells.

The ROS level in 6-OHDA-treated cells was increased up to 130% to the control level. Pre-incubation of Neuro-2a cells with the peptides at a concentration of 10 µM for 1 h before 6-OHDA addition resulted in a decreased ROS level (Figure 5). The most noticeable decrease of ROS was observed for HCRG21, a blocker of TRPV1, that reduced the ROS below the untreated control (87% to control level) obtained for cells without 6-OHDA (Figure 5). Kv blockers HCRG1 and HCRG2, which did not show significant improvement in cell viability, kept ROS at the reference level. InhVJ also kept intracellular ROS at the reference level (Figure 5).

We investigated the concentration range (0.01–10 µM) in which InhVJ influences cell viability and the ROS level in 6-OHDA-treated Neuro-2a cells (Figure 6). A direct relationship between an increase in viability (Figure 6a) and a decrease in ROS level (Figure 6b) depending on the concentration of the peptide was revealed. Increasing of InhVJ concentration caused enhancing of the anti-ROS effect, the maximal and significant antioxidant effect was observed at concentrations 1 μM and 10 μM.

### 3.4. Antioxidant Activity of Kunitz-Type Peptides

The decrease of ROS level in 6-OHDA-treated cells may be caused by direct antioxidant effect of studying peptides as well as their influence on cellular antioxidant machinery. We tested antioxidant activity of all peptides at concentration of 10 µM in DPPH radical scavenging cell free assay (Table 3). Most peptides showed a weak radical scavenging activity. Maximal antiradical effect was observed for HCRG21 and HCRG2 which scavenged 14.5% and 12.9% of DPPH radicals, respectively.

### 3.5. Effect of Kunitz-Type Peptides on TRPV1 and Kv Channels

Peptides HCGS1.10 and HCGS1.36 have similar surface electrostatic potential distribution and cluster together on a phylogenetic tree with the TRPV1 inhibitors APHC1-3 [3] and HCRG21 [10] as well as InhVJ with the latter being inactive for this channel [23]. HCGS1.19 and HCGS1.20 also share a high percentage of amino acid sequence identity with listed peptides. For this reason, we checked HCGS1.10, HCGS1.36, HCGS1.19, and HCGS1.20 in concentration 10 µM in the electrophysiological assay on TRPV1, Kv1.1, Kv1.2, Kv1.3, Kv1.4, Kv1.5, Kv1.6, Shaker IR, and hERG expressed in *Xenopus laevis* oocytes. At this concentration, no activity was observed for any of the peptides tested (Figure 7). 

## 4. Discussion

In this work, we investigated the eight Kunitz-type peptides of the sea anemone *H. crispa* in the 6-OHDA-induced neurotoxicity model in vitro. Peptides InhVJ, HCRG1, and HCRG2 are major components of the venom and were isolated from sea anemones of the *Heteractis* genus [14,22,23], while HCGS1.10, HCGS1.19, HCGS1.20, HCGS1.36, and HCRG21 were derived from transcripts [10,18] and obtained as recombinant analogs. These peptides share a sequence identity from 71 to 95% (from 3 to 10 amino acid substitutions) (Figure 8), which drives the difference in their action and results of tests in the neurotoxicity model. 

Using the example of InhVJ, we were able to carry out a detailed comparison of the properties of native and recombinant peptides. As a result, an almost complete correspondence of the structure–function parameters of the native and recombinant peptides was established. Therefore, the recombinant analog of InhVJ was used in 6-OHDA-induced neurotoxicity model. A notable fact is that the super production (yield 18.5 mg/L) was observed for recombinant InhVJ under conditions of heterologous expression and isolation, identical for all Kunitz-type peptides mentioned in the work (yield 4.27–9.38 mg/L) [10,15,24,25] regardless of whether the peptide-coding gene had been optimized for expression in *E. coli* or not. We assume that this may be due to the sequence of the InhVJ, which favors its synthesis, folding and retention in the cytoplasm of *E. coli*. The super production has practical value for fast and fruitful obtaining of the target peptide in sufficient quantities for further studies.

Parkinson’s disease (PD) is one of the most common age-related motoric neurodegenerative diseases. Pathogenesis of PD includes neuronal death as a result of oxidative stress mediated by increasing intracellular levels of ROS and reactive nitrogen species. The hyperproduction of ROS causes damage to different cell components such as DNA, lipids and proteins. Peroxidation of the latter promotes mitochondrial injury. In addition, ROS cause mitochondrial dysfunction as well as activation of apoptosis-related death signaling, resulting in neuronal cell death [44]. PD reproducing for in vivo or in vitro research is possible because a number of substances leading to specific neurodegeneration are known [43,45,46]. 6-OHDA is a neurotoxic compound able to penetrate into cells using dopamine or norepinephrine transporters [47] and initiate intracellular oxidative stress via producing ROS through enzymatic or nonenzymatic auto-oxidation [43,48,49].

It is well known that a number of naturally occurring proteins and peptides have neuroprotective properties. Peptides with neuroprotective activity can prevent neuronal cell death in a variety of ways, including inhibition of Ca^2+^ influx by blocking the DAPK1/NR2B combination, inhibition of ROS and inflammatory cytokine formation through antioxidant activity, prevent the loss of mitochondrial membrane potential and mitochondrial dysfunction, inhibition of cytochrome C release and Bax expression, increasing Bcl-2 expression, suppression of caspase pathways associated with cell death and prevention of DNA fragmentation [50]. Moreover, neuroprotective peptides of marine origin isolated from such objects as salmon, stingray, cod, trout, tilapia, tunicate and some other sea animals have been found to eliminate the toxicity of amyloid beta, suppress the expression of cAMP and caspase-3, reduce the neurotoxic effects of H_2_O_2_, ethanol and glutamate, provide anti-inflammatory effect and enhance the production of BDNF and PSD95 [51,52].

In our work, we have found that the neuroprotective properties of sea anemone *H. crispa* peptides are associated with the ability of these compounds to reduce the content of cytotoxic ROS in neuronal cells, the production of which was increased by 6-OHDA neurotoxin. It is possible that the observed effect is due to the ability of these peptides to act as free-radical scavenger. Moreover, the ROS decrease can be explained by the ability of some peptides to interact with ion channels, which, according to the published data, are involved in neurodegeneration processes.

Considering that the changes in the calcium homeostasis and oxidative stress are associated with the degeneration of dopaminergic neurons in substantia nigra, it has been suggested that TRP channels could mediate some of the mechanisms that lead to the development of the PD [53]. It has been reported that TRPV1 activation triggers Ca^2+^-dependent cell death [54] and NADPH-oxidase-mediated ROS production in microglia [55]. Such a mechanism could probably occur in neuron degeneration. Two TRPV1 antagonists, capsazepine and iodo-resiniferatoxin, could inhibit dopaminergic neurons degeneration in vivo and in vitro [56].

Recently, it has been shown, that TRPV1 channels are involved in the generation of pain which accompanies neurodegenerative diseases. Injections of 6-OHDA into the striatum of the mouse brain have significantly changed the thermal sensitivity threshold and caused mechanical allodynia inherent in Parkinson’s disease, accompanied by a significant increase in the TRPV1 expression in mice with 6-OHDA lesion [57]. In our investigation, the TRPV1 channel blocker, HCRG21 [10], has been the most effective in the experiments for decreasing ROS production and shown the significant neuroprotective action that could be indirect evidence of the TRPV1 role in neurodegeneration. It is now considered recognized that the TRPV1 inhibitors have neuroprotective properties through the regulation of calcium ions level [58]. Moreover, TRPV1 inhibition was important for the neuroprotective effect of cannabidiol in another cell Parkinson’s disease model [59]. Perhaps, blocking TRPV1 channels may prove to be a useful approach in the treatment of neurodegenerative diseases and Parkinson’s disease in particular.

Sea anemone Kunitz-type peptides can block Kv channels [60], which are known to be actively involved in the regulation of neuronal processes and considered as potential therapeutic targets for Parkinson’s disease [61,62]. Earlier we reported that HCRG1 and HCRG2 block several Kv1.x isoforms (Kv1.1, Kv1.2, Kv1.3, Kv1.6) in nM range [31], and now they showed full suppression of ROS formation in 6-OHDA-treated cells, but these peptides did not affect cell viability. They probably will be active in vivo similar to Kunitz-type toxin PcKuz3 of *P. caribaeorum* showing neuroprotective activity on 6-OHDA treated zebrafish *Danio rerio* [32].

The peptides HCGS1.10 and InhVJ were the most active in 6-OHDA-induced neurotoxicity model and their positive effect on Neuro-2a cell viability related to decreasing ROS level in these cells. It was reported that the most active peptide InhVJ did not have any effect on Kv1.x and TRPV1 channels [23] as that was established for its homologues with point substitutions (Figure 8). InhVJ has been shown to be a specific inhibitor of trypsin and α-chymotrypsin, and it did not interact with kallikrein, thrombin, plasmin, papain, and pepsin [23,63]. Therefore, the question about additional cellular targets, which would explain the neuroprotective properties of this peptide, remains open. Peptide HCGS1.10, as well as less active peptides HCGS1.36 and HCGS1.20 also did not have any effect on Kv1.x and TRPV1 channels and the targets of their action remain to be clarified too.

The permeability of the blood–brain barrier (BBB) is one of the main factors limiting the use of pharmaceutical agents for the treatment of central nervous system diseases including Parkinson’s disease. Kunitz domain in β-amyloid precursor protein is important for transport across the BBB via low-density lipoprotein receptor-related protein (LRP) [64]. Kunitz-type peptide BPTI from *Bos taurus* (Figure 8) effectively crosses the BBB by receptor-mediated transport through LRP [65]. It was found that the C-terminal fragment (from 31 to 50 a.a.) of BPTI is responsible for penetration through the BBB. Based on this sequence, short peptides (Angiopeps ~2,3 kDa) which penetrate the BBB ten times more efficiently than BPTI were created [65]. The possibilities of their use in the composition of complex nanoparticles for the delivery of chemotherapeutic drugs to brain tumors are currently being investigated [66,67,68]. The indicated C-terminal fragment of BPTI or Angiopep-2 shares ~50% of identity with the corresponding fragment of Kunitz-type peptides from sea anemone *H. crispa*. Thus, these peptides may be able to penetrate the BBB and exhibit a cytoprotective effect in CNS. However, this assumption requires further verification in experiments on laboratory animals.

## Figures and Tables

**Figure 1 biomedicines-09-00283-f001:**
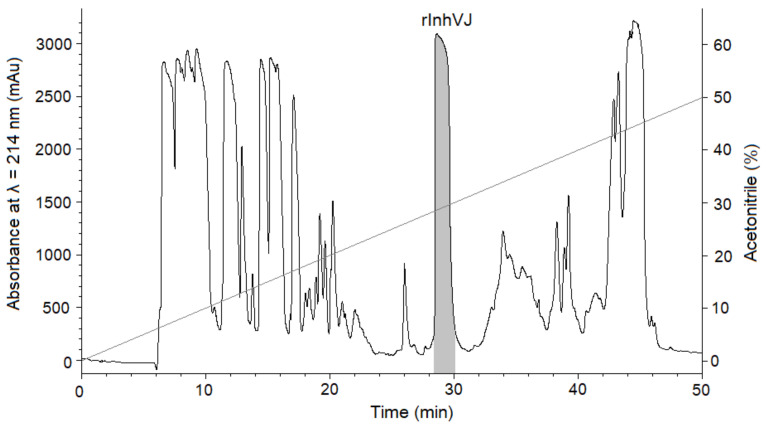
Elution profile of peptides after BrCN digestion of recombinant IhnVJ fusion protein on Jupiter C_4_ column (Phenomenex, USA) in a linear gradient of buffer B (0–50%) at a flow rate of 3 mL/min for 50 min. The fraction containing the InhVJ is filled with a light grey color.

**Figure 2 biomedicines-09-00283-f002:**
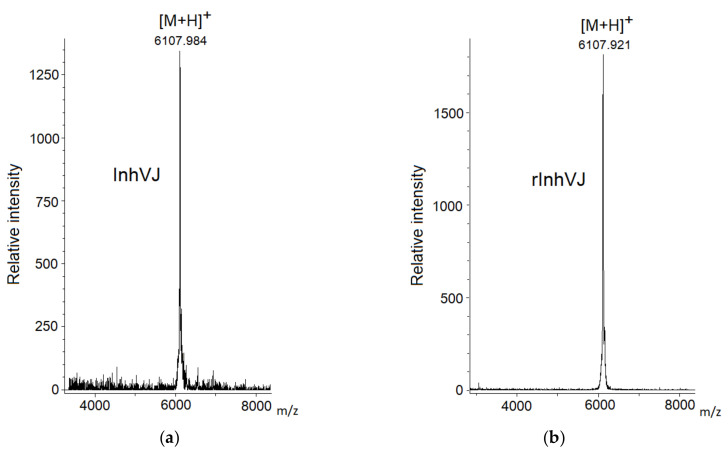
Mass spectra of native InhVJ (**a**) and recombinant InhVJ (**b**) peptides.

**Figure 3 biomedicines-09-00283-f003:**
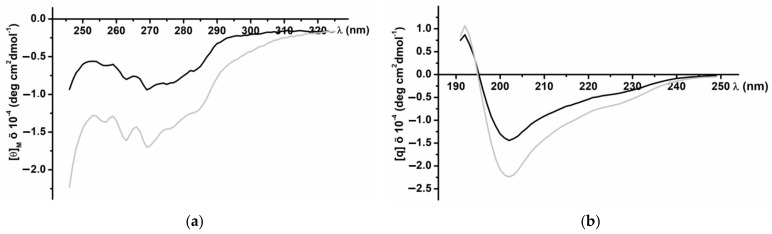
CD spectra of native (black line) and recombinant (grey line) InhVJ in 0.01M phosphate buffer, pH 7.0, in near or aromatic (**a**) and far or peptide bond (**b**) UV regions.

**Figure 4 biomedicines-09-00283-f004:**
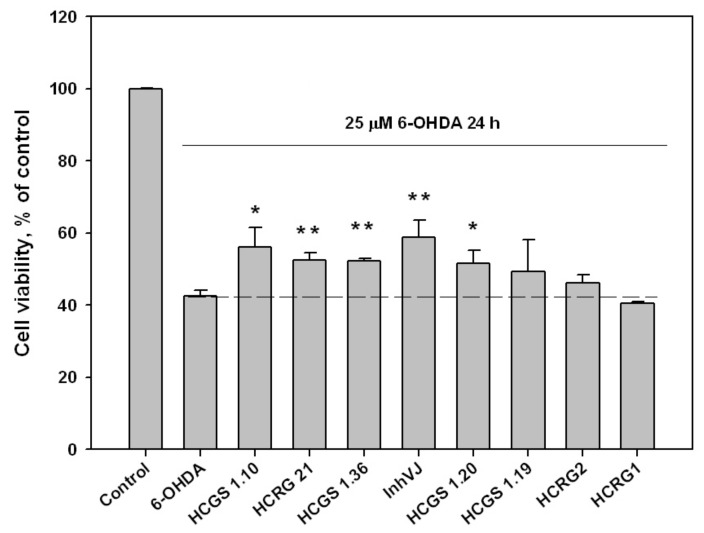
Effects of 10 μM of Kunitz-type peptides on Neuro-2a cell viability in 6-OHDA-induced neurotoxicity model. Differences between 6-OHDA-treated cells alone (designated by the dotted line) and with peptides preincubated during 1 h are significant with * *p* < 0.05 and ** *p* < 0.01. Data were obtained in three independent replicates and calculated values were expressed as mean ± SEM.

**Figure 5 biomedicines-09-00283-f005:**
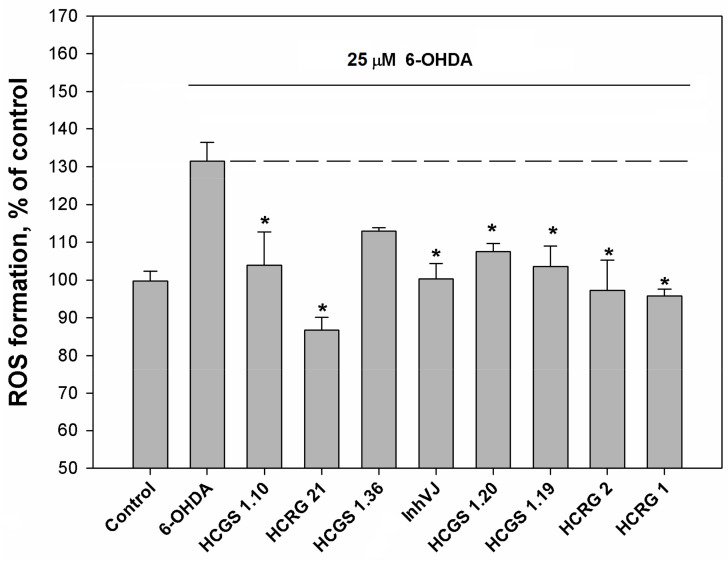
Effects of Kunitz-type peptides on ROS formation in Neuro-2a cells induced by 6-OHDA. Differences between 6-OHDA-treated cells alone (designated by the dotted line) and with peptides pre-incubated during 1 h are significant with * *p* < 0.05. Data were obtained in three independent replicates and calculated values were expressed as mean ± SEM.

**Figure 6 biomedicines-09-00283-f006:**
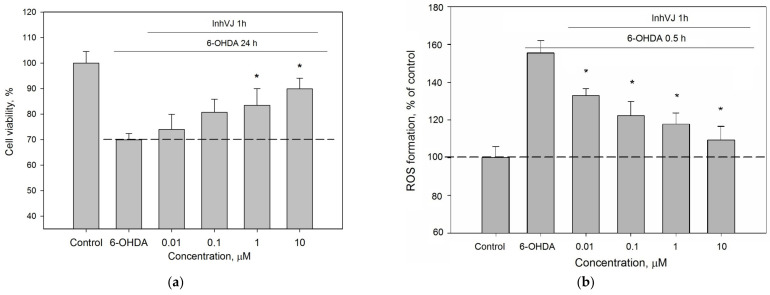
Concentration-dependent influence of InhVJ on (**a**) Neuro-2a cell viability after 24 h incubation with 25 µM of 6-OHDA. (**b**) ROS formation in Neuro-2a cells induced by 6-OHDA during 0.5 h. Each bar represents the mean ± SEM of three independent replicates. Differences between 6-OHDA-treated cells incubated alone (designated by the dotted line) or with InhVJ are statistically significant with * *p* < 0.05.

**Figure 7 biomedicines-09-00283-f007:**
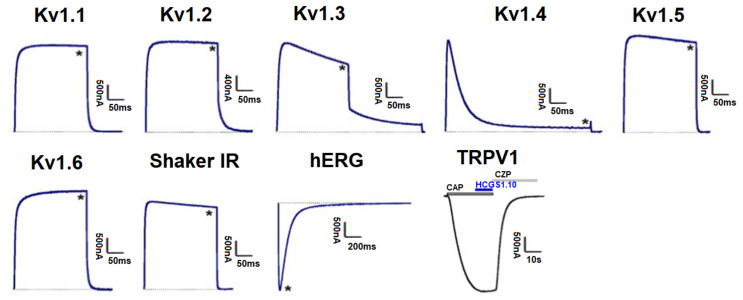
Differential effects of HCGS1.10, HCGS1.36, HCGS1.19, and HCGS1.20 on Kv isoforms and TRPV1 expressed in *Xenopus laevis* oocytes. Representative whole cell K^+^ current traces of oocytes expressing cloned Kv isoforms Kv1.1-Kv1.6, hERG (Kv11.1) and the Drosophila channel Shaker IR. The dotted line indicates zero-current level. The asterisk (*) indicates the steady-state current peak amplitude in the presence of 10 µM HCGS1.10, HCGS1.36, HCGS1.19, and HCGS1.20. Whole-cell current traces in HCGS1.10 conditions co-applied with capsaicin (CAP = 2 µM); capsazepine (CZP = 10 µM) are shown. The image was carried out by pClamp Clampfit 10.0 (Molecular Devices, Downingtown, PA, USA) and Origin 7.5 software (Originlab, Northampton, MA, USA).

**Figure 8 biomedicines-09-00283-f008:**
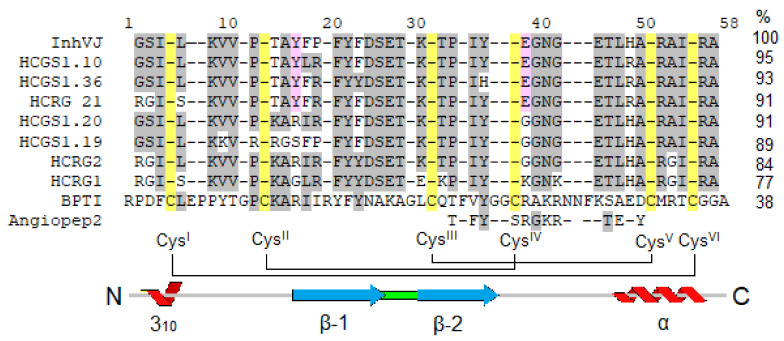
Multiple alignment of sea anemones Kunitz-type peptides using in research and BPTI from *Bos taurus* and Angiopep2 derived from it. Identical residues are shown by dashes, conservative residues are shown on light grey color, residues distinguishing active peptides are shown on pink, Cys are shown on yellow. Secondary structure elements and disulfide bond diagram are displayed according to BPTI data.

**Table 1 biomedicines-09-00283-t001:** Molecular weighs and inhibition constants of the peptides.

№	Peptide	MW Calculated, Da	MW Measured by MS, Da	K_i_ Trypsin, M
1	HCRG1	6195.6	6196 [14]	2.8 × 10^−8^ [14]
2	HCRG2	6150.6	6148 [14]	5.0 × 10^−8^ [14]
3	HCGS1.19	6087.5	6188.7	3.0 × 10^−8^ [15]
4	HCGS1.20	6079.6	6078.9	2.1 × 10^−8^ [24]
5	HCGS1.36	6174.5	6174.8	1.0 × 10^−7^ [15]
6	HCGS1.10	6150.5	6150.5	2.1 × 10^−7^ [15]
7	HCRG21	6227.5	6228.5	1.0 × 10^−7^ [10]
8	InhVJ	6106.4	6107.9	7.8 × 10^−8^

**Table 2 biomedicines-09-00283-t002:** The content of the secondary structure elements of native and recombinant InhVJ, %.

InhVJ	Helixes			β-Sheet	Turns			Others
	α	3_10_	Sum		β-Turn	PP2	Sum	
Recombinant	12.4	8.7	21.1	10.1	18.0	6.5	24.5	44.3
Native	6.9	8.8	15.7	16.3	15.7	9.3	25.0	43.0

**Table 3 biomedicines-09-00283-t003:** DPPH radical scavenging activity of Kunitz-type peptides.

Compounds, 10 µM	Scavenging of DPPH Radicals, %	Compounds, 10 µM	Scavenging of DPPH Radicals, %
HCGS1.10	8.9 ± 3.4 *	HCGS 1.20	11.7 ± 2.4 *
HCRG21	14.5 ± 4.1 *	HCGS 1.19	2.4 ± 1.0
HCGS1.36	6.2 ± 2.6	HCRG2	12.9 ± 3.0 *
InhVJ	8.1 ± 1.9 *	HCRG1	10.9 ± 3.5 *
Ascorbic acid	33.8 ± 2.4		

Differences between peptide scavenging effect and control are significant with * *p* < 0.05. Data were obtained in three independent replicates and calculated values were expressed as mean ± SEM.

## Data Availability

Data available in a publicly accessible repository.

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
