# Peer review of "Sea Anemone Kunitz-Type Peptides Demonstrate Neuroprotective Activity in the 6-Hydroxydopamine Induced Neurotoxicity Model"

_biomedicines, 2021, doi:10.3390/biomedicines9030283_

Round 1
Reviewer 1 Report
Reviewer’s comments (Biomedicines-1115433):
The manuscript by Sintsova and collaborators focuses on eight Kunitz-type peptides (water extracted or recombinantly expressed) from the sea anemone Heteractis crispa. In this study, the authors are examining molecules with some cytoprotective potential in the 6-hydroxydopamine (6-OHDA)-induced Parkinson's disease model on neuroblastoma (Neuro-2a) cells. They found that five peptides are increasing the viability of 6-OHDA-treated neuronal cells. HCRG21 (TRPV1 channel blocker) also showed some significant neuroprotection, suggesting the contribution of TRPV1 channel in the neurodegenerative disorders. Pre-treatment of neuroblastoma cells with HCRG21, followed by 6-OHDA treatment, resulted in a marked reduction in reactive oxygen species (ROS) production (similar to the untreated cells). The ‘lead’ peptide of this study, referred to as InhVJ, showed a neuroprotective potential in the micromolar concentration range.
In my opinion, this is an interesting and (overall) scientifically sound work. The study provides some important information on the potential neuroprotective effects of sea anemone Kunitz-type peptides in the 6-OHDA-induced Parkinson’s disease model on neuroblastoma (Neuro-2a) cells. It is also worth mentioning that the manuscript is clear, well-written and reasonably discussed. Illustrations of good quality appear to be appropriate.
I have a few main concerns to potentially improve the quality of the manuscript, as follows:
- One may regret that the cellular target(s) of InhVJ, as well as of the other bioactive peptides, has (have) not been formally identified/characterized;
- Did the authors investigate the antioxidant potentials of the tested peptides?
Indeed, according to the primary structures, these particular peptides may well possess some antioxidant properties impacting ROS levels in their experiments;
- It is stated that the TRPV1 channel blocker HCRG21 revealed the neuroprotective effect confirming TRPV1 involvement in the disorders associated with neurodegeneration. Could HCRG21 be also active on another ion channel (or distinct target) that would be actually responsible of the observed neuroprotective effect?
- Strictly, does the experimental approach used by the authors confirm the TRPV1 involvement in the disorders associated with neurodegeneration? The authors should argue on this point.
Minor points are as follows:
- Figure 1: The ordinate legend ‘Absorbance at 214 nm (mAu)’ should be replaced by ‘Absorbance at λ = 214 nm (mAu)’;
- Figure 2, (a) and (b): The ‘% Intencity’ should be replaced by ‘Relative intensity’ or ‘Number of counts’ (corresponding to the number of ionized molecules (here (M+H)+) hitting the detector in MALDI-TOF mass spectrometry). Also, ‘(M+H)+’ should be noted aside the peak, or in the legend of figure;
- Figure S1:
(a) The legend ‘Absorbance at 214 nm’ should be replaced by ‘Absorbance at λ = 214 nm’;
(b) The ‘% Intencity’ should be replaced by ‘Relative intensity’ or ‘Number of counts’ (corresponding to the number of ionized molecules (here (M+H)+) hitting the detector in MALDI-TOF mass spectrometry). Also, ‘(M+H)+’ should be noted aside the peak, or in the legend of figure;
- Figure S2: same comments as those reported in Figure S1 (b).
Author Response
G.B. Elyakov Pacific Institute of Bioorganic Chemistry
Far Eastern Branch, Russian Academy of Sciences
159 100-let Vladivostoku Ave., Vladivostok, 690022, Russia. ( 7(423) 231-14-30;
fax: 7(423) 231-40-50; e-mail: piboc@eastnet.febras.ru
To: Reviewer
February 26, 2021
Dear Reviewer,
Thank you for your comprehensive evaluation of our manuscript. We carefully read the review and tried to take into account all your comments in the revised manuscript. The responses to your comments are given below.
- “One may regret that the cellular target(s) of InhVJ, as well as of the other bioactive peptides, has (have) not been formally identified/characterized;”
We will continue to search targets both for InhVJ and other peptides as soon as we have the opportunity to expand the range of targets available to us for research.
- “Did the authors investigate the antioxidant potentials of the tested peptides? Indeed, according to the primary structures, these particular peptides may well possess some antioxidant properties impacting ROS levels in their experiments;”
Thank you so much for the offer to test antioxidant potential of the peptides. We have tested antioxidant activity of the peptides by DPPH methods and supplemented the corresponding sections with the obtained results (marked in blue)
In Materials and Methods:
2.11. DPPH radical scavenging assay
The 2,2-diphenyl-1-picryl-hydrazyl-hydrate (DPPH) radical scavenging activity was tested as described previously [DOI: 10.3390/md16110457].The compound solutions (120 µL) were dispensed into wells of a 96-well microplate. The DPPH (Sigma-Aldrich, Germany) was dissolved in MeOH at concentration of 7.5×10-3 M and the solution (30 µL) was added to each well. The concentrations of tested compounds in the mixtures were 10 µM. The mixtures were shaken and left for 30 min. The absorbance of the resulting solutions was measured at λ = 520 nm with a MultiscanFC plate reader (ThermoScientific, USA). The scavenging of the DPPH radical in comparison with control (MeOH) (%) was calculated for each compound. Ascorbic acid at 10 µM was used as positive control.
In Results:
3.4. Antioxidant activity of Kunitz-type peptides
The decrease of ROS level in 6-OHDA-treated cells may be caused by direct antioxidant effect of studying peptides as well as their influence on cellular antioxidant machinery. We tested antioxidant activity of all peptides at concentration of 10 µM in DPPH radical scavenging cell free assay (Table 3). Most peptides showed a weak statistical radical scavenging activity. Maximal antiradical effect was observed for HCRG21 and HCRG2 which scavenged 14.5% and 12.9% of DPPH radicals, respectively.
Table 3. DPPH radical scavenging activity of Kunitz-type peptides
|
Compounds, 10 µM |
Scavenging of DPPH radicals, % |
Compounds, 10 µM |
Scavenging of DPPH radicals, % |
|
HCGS1.10 |
8.9 ± 3.4 * |
HCGS 1.20 |
11.7 ± 2.4 * |
|
HCRG21 |
14.5 ± 4.1 * |
HCGS 1.19 |
2.4 ± 1.0 |
|
HCGS1.36 |
6.2 ± 2.6 |
HCRG2 |
12.9 ± 3.0 * |
|
InhVJ |
8.1 ± 1.9 * |
HCRG1 |
10.9 ± 3.5 * |
|
Ascorbic acid |
33.8 ± 2.4 |
|
|
Differences between peptide scavenging effect and control are significant with * p < 0.05. Data were obtained in three independent replicates and calculated values were expressed as mean ± SEM.
In Discussion:
It is possible that the observed effect is due to the ability of these peptides act as an efficient free-radical scavenger.
- “It is stated that the TRPV1 channel blocker HCRG21 revealed the neuroprotective effect confirming TRPV1 involvement in the disorders associated with neurodegeneration. Could HCRG21 be also active on another ion channel (or distinct target) that would be actually responsible of the observed neuroprotective effect?”
According to our previous study (Monastyrnaya et al., 2016) TRPV1 blocker HCRG21 is not active on Kv isoforms (Kv1.1–Kv1.6, hERG, and Shaker IR). We agree with you, that HCRG21 can be also active on another ion channel or distinct target that would be actually responsible of the observed neuroprotective effect.
We added new paragraphs in Discussion (marked in blue):
It is well known that a number of naturally occurring proteins and peptides have neuroprotective properties. Peptides with neuroprotective activity can prevent neuronal cell death in a variety of ways, including inhibition of Ca2+ influx by blocking the DAPK1/NR2B combination, inhibition of ROS and inflammatory cytokine formation through antioxidant activity, prevent the loss of mitochondrial membrane potential and mitochondrial dysfunction, inhibition of cytochrome C release and Bax expression, increasing Bcl-2 expression, suppression of caspase pathways associated with cell death and prevention of DNA fragmentation [Lee S.Y., 2019]. Moreover, neuroprotective peptides of marine origin isolated from such objects as salmon, stingray, cod, trout, tilapia, tunicate and some other sea animals were found to eliminate the toxicity of amyloid beta, suppress the expression of cAMP and caspase-3, reduce the neurotoxic effects of H2O2, ethanol and glutamate, provide anti-inflammatory effect and enhance the production of BDNF and PSD95 [Pangestuti R., 2013; Ryu B., 2013].
In our work, we found that the neuroprotective properties of sea anemone H. crispa peptides are associated with the ability of these compounds to reduce the content of cytotoxic ROS in neuronal cells, the production of which was increased by 6-OHDA neurotoxin. It is possible that the observed effect is due to the ability of these peptides act as an efficient free-radical scavenger. Moreover, the ROS decrease can be explained the ability of some peptides to interact with ion channels, which, according to the published data, are involved in neurodegeneration processes.
- “Strictly, does the experimental approach used by the authors confirm the TRPV1 involvement in the disorders associated with neurodegeneration? The authors should argue on this point”.
We completely agree with you that the experimental approach used by us doesn’t confirm the TRPV1 involvement in the disorders associated with neurodegeneration. Of course, for direct proof, additional experiments are needed for example using a cell line with knockout genes of TRPV1.
We rephrased sentences and added new paragraphs in the manuscript (all change in the manuscript marked in blue)
in Abstract:
The TRPV1 channel blocker, HCRG21, revealed the neuroprotective effect that could be indirect evidence of TRPV1 involvement in the disorders associated with neurodegeneration.
in Discussion:
Considering that calcium homeostasis changing and oxidative stress are associated with the degeneration of dopaminergic neurons (DN) in substantia nigra, it has been suggested that TRP channels could mediate some of the mechanisms that lead to the development of the PD. It has been reported, that TRPV1 activation triggers Ca2+-dependent cell death (Kim et al., 2006) and NADPH-oxidase-mediated ROS production in microglia (Shirakawa and Kaneko, 2018). Such mechanism could probably occur in neuron degeneration. Two TRPV1 antagonists, capsazepine and iodo-resiniferatoxin, could inhibit DN degeneration in vivo and in vitro (Kim et al., 2005).
Recently it has been shown TRPV1 channels are involved in the generation of pain accompanying neurodegenerative diseases. Injection of 6-OHDA into the striatum of the mouse brain significantly changes the thermal sensitivity threshold and causes mechanical allodynia inherent in Parkinson's disease, accompanied by a significant increase in TRPV1 expression in mice with 6-OHDA lesion (Li M. et al., 2020). In our investigation, the TRPV1 channel blocker, HCRG21 [10], was the most effective in experiments for decreasing ROS production and showed the significant neuroprotective action that could be indirect evidence of TRPV1 role in neurodegeneration. It is now considered recognized that the TRPV1 inhibitors have neuroprotective properties through the regulation of calcium ions level [51]. Moreover, TRPV1 inhibition was important for the neuroprotective effect of cannabidiol in another cell neurotoxicity model [52]. Perhaps, blocking TRPV1 channels may prove to be a useful approach in the treatment of neurodegenerative diseases and Parkinson's disease in particular.
- “Figure 1: The ordinate legend ‘Absorbance at 214 nm (mAu)’ should be replaced by ‘Absorbance at λ = 214 nm (mAu)’;”
We added “λ =”
- “Figure 2, (a) and (b): The ‘% Intencity’ should be replaced by ‘Relative intensity’ or ‘Number of counts’ (corresponding to the number of ionized molecules (here (M+H)+) hitting the detector in MALDI-TOF mass spectrometry). Also, ‘(M+H)+’ should be noted aside the peak, or in the legend of figure;”
We replaced ‘% Intencity’ by ‘Relative intensity’
- “Figure S1:
(a) The legend ‘Absorbance at 214 nm’ should be replaced by ‘Absorbance at λ = 214 nm’;
(b) The ‘% Intencity’ should be replaced by ‘Relative intensity’ or ‘Number of counts’ (corresponding to the number of ionized molecules (here (M+H)+) hitting the detector in MALDI-TOF mass spectrometry). Also, ‘(M+H)+’ should be noted aside the peak, or in the legend of figure;”
We replaced ‘Absorbance at 214 nm’ by ‘Absorbance at λ = 214 nm’ and ‘% Intencity’ by ‘Relative intensity’ and noted ‘(M+H)+’ aside the peak
- “Figure S2: same comments as those reported in Figure S1 (b)”.
We replaced ‘% Intencity’ by ‘Relative intensity’ and noted ‘(M+H)+’ aside the peak
In addition, according to your recommendation, we have improved the style and grammar of English.
We hope that you are satisfied with our responses and you will give a positive decision on our manuscript.
Best regards,
Elena Leychenko
G.B. Elyakov Pacific Institute of Bioorganic Chemistry,
Far Eastern Branch of the Russian Academy of Sciences,
Prospect 100-let Vladivostoku, 159,
690022, Vladivostok, Russia
E-Mail: leychenko@gmail.com
Reviewer 2 Report
Major comments:
- Could the peptides cross the blood-brain barrier? Could the authors give direct evidence?
- The authors only tested the cell viability in their study, this in vitro experiment cannot be used as an effective model of PD, given PD is a heterogenous disease with a varying age of onset, symptoms, and rate of progression. The title of this article is not accurate.
Author Response
G.B. Elyakov Pacific Institute of Bioorganic Chemistry
Far Eastern Branch, Russian Academy of Sciences
159 100-let Vladivostoku Ave., Vladivostok, 690022, Russia. ( 7(423) 231-14-30;
fax: 7(423) 231-40-50; e-mail: piboc@eastnet.febras.ru
To: Reviewer
February 27, 2021
Dear Reviewer,
Thank you for your comprehensive evaluation of our manuscript. We carefully read the review and tried to take into account all your comments in the revised manuscript. The responses to your comments are given below.
- “Could the peptides cross the blood-brain barrier? Could the authors give direct evidence?”
To date, we have only indirect evidence that the Kunitz-type peptides of sea anemones can pass through the BBB. We believe that by analogy with BPTI from Bos taurus, for which penetration through the BBB has been established, peptides with a similar fold will also be able to pass through the BBB. However, we plan to resolve this important issue as soon as possible since we are also interested in similar experiments. Now we are screening and selecting the most effective representatives of Kunitz peptides in in vitro models, then we are going to test the peptides in in vivo models of PD and we are planning to carry out experiments that will allow us to answer the question if the peptides pass through the BBB.
- “The authors only tested the cell viability in their study, this in vitro experiment cannot be used as an effective model of PD, given PD is a heterogenous disease with a varying age of onset, symptoms, and rate of progression. The title of this article is not accurate.”
We completely agree with your remark and made corrections both in the title of the article: “Sea anemone Kunitz-type peptides demonstrate neuroprotective activity in the 6-hydroxydopamine induced neurotoxicity model” and in the text (marked in red).
In the abstract:
We have studied eight Kunitz-type peptides of the sea anemone Heteractis crispa to search molecules with cytoprotective activity in the 6-OHDA-induced neurotoxicity model on neuroblastoma Neuro-2a cells.
The target of InhVJ is still unknown, but it was the best of all eight homologous peptides in an absolute cell viability increment on 38% of the control in the 6-OHDA-induced neurotoxicity model.
In the chapter 3.3:
We examined the effect of peptides set on cell viability in 6-OHDA-induced neurotoxicity model.
It is believed that the cell death in the 6-OHDA-induced neurotoxicity model is a result of dramatically increased reactive oxygen species (ROS) content.
In the legend of Fig.4:
Figure 4. Effects of 10 μM of Kunitz-type peptides on Neuro-2a cell viability in 6-OHDA-induced neurotoxicity model.
In the chapter 4:
These peptides share a sequence identity from 71 to 95% (from 3 to 10 amino acid substitutions) (Figure 8), which drives the difference in their action and results of tests in the neurotoxicity model.
Therefore, the recombinant analog of InhVJ was used in 6-OHDA-induced neurotoxicity model.
Moreover, TRPV1 inhibition was important for the neuroprotective effect of cannabidiol in another cell neurotoxicity model.
The peptides HCGS1.10 and InhVJ were the most active in 6-OHDA-induced neurotoxicity model and their positive effect on Neuro-2a cell viability related to decreasing ROS level in these cells.
In addition, according to your recommendation, we have worked on the manuscript and improved the style and grammar of English
We hope that you are satisfied with our responses and you will give a positive decision on our manuscript.
Best regards,
Elena Leychenko
G.B. Elyakov Pacific Institute of Bioorganic Chemistry,
Far Eastern Branch of the Russian Academy of Sciences,
Prospect 100-let Vladivostoku, 159,
690022, Vladivostok, Russia
E-Mail: leychenko@gmail.com
Round 2
Reviewer 1 Report
In my opinion, this revised manuscript could be accepted for publication in its present form.
Reviewer 2 Report
The authors adequately answered my concerns.